# Lung Cancer Incidence by Detailed Race–Ethnicity

**DOI:** 10.3390/cancers15072164

**Published:** 2023-04-05

**Authors:** Hannah M. Cranford, Tulay Koru-Sengul, Gilberto Lopes, Paulo S. Pinheiro

**Affiliations:** 1Department of Public Health Sciences, University of Miami Miller School of Medicine, Miami, FL 33136, USA; 2Sylvester Comprehensive Cancer Center, University of Miami Miller School of Medicine, Miami, FL 33136, USA; 3Department of Medicine, University of Miami Miller School of Medicine, Miami, FL 33136, USA

**Keywords:** lung cancer, incidence, risk, rate, race/ethnicity, Hispanic, Black, Cubans, Mexicans, Puerto Rican, Central American, South American, Caribbean

## Abstract

**Simple Summary:**

Lung cancer incidence patterns and clinical characteristics across the heterogeneous non-Hispanic Black and Hispanic racial–ethnic populations of the United States (US) are understudied. This characterization of lung cancer rates across US minority populations is important for targeting clinical and public health measures in high-risk populations. The aim of our population-based study is to assess lung cancer heterogeneity among these populations by detailed race–ethnicity or nativity (e.g., Cubans, Mexicans, Caribbean-born Blacks) using all lung cancer cases from the Florida state cancer registry, 2012–2018, and computing, for the first time, age-adjusted incidence rates (AAIR) for each population. AAIRs among Blacks and Hispanics vary approximately 3-fold between detailed groups and highlight the need to look beyond aggregate groups for tailored approaches in the fight against lung cancer. The implications of these findings are significant for public health surveillance and for clinical professionals working with diverse US populations.

**Abstract:**

Lung cancer (LC) incidence rates and tumor characteristics among (non-Hispanic) Black and Hispanic detailed groups, normally characterized in aggregate, have been overlooked in the US. We used LC data from the Florida state cancer registry, 2012–2018, to compute LC age-adjusted incidence rates (AAIR) for US-born Black, Caribbean-born Black, Mexican, Puerto Rican, Cuban, Dominican, and Central and South American populations. We analyzed 120,550 total LC cases. Among Hispanics, Cuban males had the highest AAIR (65.6 per 100,000; 95%CI: 63.6–67.6), only 8% [Incidence Rate Ratio (IRR): 0.92; 95%CI: 0.89–0.95] lower than Whites, but 2.7 (IRR 95%CI: 2.31-3.19) times higher than Central Americans. Among Blacks, the AAIR for US-born Black males was over three times that of those Caribbean-born (IRR: 3.12; 95%CI: 2.80–3.40) and 14% higher than White males (IRR: 1.14; 95%CI: 1.11–1.18). Among women, US-born Blacks (46.4 per 100,000) and foreign-born Mexicans (12.2 per 100,000) had the highest and lowest rates. Aggregation of non-Hispanic Blacks or Hispanics obscures inherent disparities within groups. Understanding the distinct LC rates in US populations is crucial for targeting public health measures for LC diagnosis, prevention, and treatment. Further LC research exploring detailed race–ethnicity regarding LC in never-smokers is necessary, particularly among females and considering pertinent environmental factors.

## 1. Introduction

Lung cancer in the United States of America (US) has the highest overall mortality rate and is the second in overall incidence, compared to other cancers [1,2]. Despite declines for all racial–ethnic groups, largely due to a declining prevalence in cigarette smoking, there still exist considerable disparities by race–ethnicity. During 2014–2018, non-Hispanic Black males maintained the highest rates of lung cancer incidence and mortality among all race–ethnicities [1], and among Hispanics, lung cancer was the leading cause of cancer death for males and second for females, after breast cancer [1,3]. However, both (non-Hispanic) Black populations (e.g., US-born Blacks, Caribbean-born Blacks, African Immigrants) and Hispanic populations (e.g., Puerto Ricans, Cubans, Mexicans, etc.) are vastly heterogeneous, representing distinct nativity, language dominance, duration of US residence, socioeconomic status, and behavioral risk factors, which reflect distinctly in stratified cancer incidence and mortality among detailed groups [4,5,6,7]. However, few studies have explored lung cancer incidence patterns [6] in such detailed racial–ethnic groups and their inherent characteristics, such as sex, age group, socioeconomic status, histology, and stage at diagnosis.

In existing population-based studies, there is little recognition of the diversity within the US Black population because US cancer surveillance program, the North American Association of Central Cancer Registries (NAACCR), standards do not collect information on heterogeneity among this racial group. As a result, most studies categorize Blacks based on nativity and are majority mortality studies [8,9], because place of birth is nearly complete via death certificate documentation. Incidence studies are rarer, as incomplete knowledge on birthplace is problematic and a source of considerable bias in population rates [10], a problem that extends to incidence studies on Hispanic groups because the collected ethnic group variable is often equally incomplete [5,10,11]. As the US population continues to diversify, the characterization of lung cancer incidence rates by Black and Hispanic detailed race–ethnicity is critical for furthering public health advances in cancer diagnosis, treatment, and prevention tailored to distinct groups with varying smoking prevalence, environmental exposures, and other lung cancer risk factors [4,12,13,14,15].

To further the discourse regarding granular analysis of US Black and Hispanic populations and to address the lack of specific lung cancer incidence rates and characteristics by detailed racial–ethnic group, we conducted a population-based analysis of lung cancer incidence among major racial–ethnic groups and specific Black and Hispanic groups in the diverse state of Florida, the third US state in population and the second US state in number of cancer cases, with over 150,000 newly diagnosed cancers in 2022 [16].

## 2. Materials and Methods

Individual level data from all lung cancer cases (International Classification of Diseases (ICD): ICD-O-3 codes C34.X and histology codes 8000–9540) diagnosed during 2012–2018 were obtained from the Florida Cancer Data System (FCDS), including demographic, tumor, and socioeconomic characteristics. FCDS is the legislatively mandated, population-based central cancer registry for Florida, which has been collecting incidence data from hospitals, radiation centers, surgery centers, and physician offices since 1981. FCDS has met or exceeded the NAACCR standards of quality, timeliness, and completeness (>95%) for every year since 1995.

Data were aggregated on race and ethnicity and were reported in four major, mutually exclusive, racial–ethnic groups [non-Hispanic White (White), non-Hispanic Black (Black), non-Hispanic Asian and Pacific Islander (API), and Hispanic] and 11 detailed racial–ethnic groups: among Blacks, US-born and Caribbean-born; among Hispanics, Mexican, with two divisions (US-born Mexican, foreign-born Mexican), Puerto Rican, Cuban, Dominican, Central American, South American, and combined Central and South American.

Gathered FCDS characteristics comprise sex, age at diagnosis, histology, microscopic verification, stage at diagnosis, socio-economic status, and insurance type. Sex as female include n = 41 identified as non-binary (fewer than 0.07% of total females). Histology is classified according to previous research [17] into the following types: adenocarcinoma (8050, 8140–8147, 8201, 8250–8255, 8260, 8290, 8310, 8320, 8323, 8333, 8470, 8480–8490, 8507, 8550, 8570–8574, 8576), squamous cell carcinoma (8052, 8070–8076, 8083, 8084, 8120–8123), non-small cell lung cancer not otherwise specified (NOS) (8046), small cell carcinoma (8002, 8041–8045), large cell carcinoma (8012–8014, 8021, 8082), other specified (8003–8005, 8022, 8023, 8030–8035, 8170, 8200, 8240–8249, 8256, 8257, 8265, 8340, 8430, 8551, 8560, 8562, 8575, 8640, 8720, 8772, 8800–8815, 8830, 8842, 8850–8854, 8890–8901, 8912, 8940, 8963, 8972, 8980, 9041, 9065, 9085, 9100, 9120, 9130, 9150, 9181, 9231, 9540), and unspecified (8000, 8001, 8010, 8020, 8230). Histologic code 8000 comprises cases without microscopic verification. The Surveillance, Epidemiology, and End Results (SEER) [18] staging categories (localized, regional, distant, and unknown) were used to identify cancer stage at diagnosis. Census tract poverty level was used as a measure of socio-economic status and was defined as the proportion of the population living below census tract poverty level with categorization as follows: very low (0% to < 5%), low (5% to < 10%), medium (10% to < 20%), high (20% to < 100%), or unknown. Insurance type was classified as Medicaid, Medicare, private, no insurance, or unknown.

Demographic and tumor characteristics were compared among the four major racial–ethnic groups (White, Black, API, and Hispanic), among Blacks (US-born and Caribbean-born), among Hispanics (Mexican, Puerto Rican, Cuban, Dominican, Central American, and South American), and among Mexicans (US-born and foreign-born) by Chi-square tests of differences: 5-year age group, histology, microscopic verification, SEER stage at diagnosis, census tract poverty level, and insurance status.

Missingness among detailed racial–ethnic group was 20.1% of Hispanic cases (n = 2427 of 12,062) and 13.9% of non-Hispanic Black cases (n = 1346 of 9710). To avoid underestimating rates by racial–ethnic group status and obtain comparable numerators for our population-based rates, we conducted multiple imputations of specific ethnic (for Hispanics) and nativity group (for Blacks), with 20 iterations based on a multivariate imputation by fully specification method [19,20] for each racial–ethnic group (i.e., Hispanics and non-Hispanic Blacks), and took into account joint distribution for variables: sex, smoking status at diagnosis, 5-year age group, histology, Florida region of residence, year of diagnosis, and nativity (for Hispanic group only). For this cross-sectional analysis of lung cancer, detailed population denominators for each Florida racial–ethnic group were obtained from the US Census Bureau’s single-year American Community Survey data, pooled for a 7-year period (2012–2018) [21].

Lung cancer incidence rates by detailed race–ethnicity and sex were calculated per 100,000 people and age-adjusted by the direct method to the 2000 US standard population. Tiwari incidence rate ratios (IRR) with 95% confidence intervals (CI) [22] were computed and are expressed with non-Hispanic White as reference, except when otherwise mentioned. American Indian and Alaska Native populations were excluded from incidence calculations due to sparse data.

SAS v9.4 software (SAS Institute Inc., Cary, NC, USA) was used for data management and statistical analysis. Exempt status was granted by the Florida Department of Health’s Institutional Review Board.

## 3. Results

Of 120,550 lung cancer cases diagnosed in Florida during 2012–2018, the majority identified as White (79.7%; n = 96,131), with 51.0% males (n = 49,073) and 49.0% females (n = 47,088); following were Hispanics at 10.0% (n = 12,062) with 58.3% (n = 7038) and 41.7% (n = 5024) males and females, respectively; Blacks at 8.1% (n = 9710) with 57.2% (n = 5553) and 42.8% (n = 4157) males and females, respectively; and API at 0.9% (n = 1030) with 49.7% (n = 512) and 50.3% (n = 518) males and females, respectively (Table 1). Cubans (55.4%; n = 6680) and Puerto Ricans (18.7%; n = 2250) comprised most Hispanic lung cancer cases (Table 2), while cases in Black patients rested largely among those who were US-born (87.0%; n = 8452) versus Caribbean-born (12.0%; n = 1169).

Compared to other Hispanic groups, Cuban and foreign-born Mexican patients disproportionately reside in areas of high poverty (44.2% and 45.2% high poverty for males and 39.9% and 43.3% for females, respectively). There are also significant differences in Hispanic lung cancer patients by insurance status, with Mexicans and Cubans having lower proportions of private insurance (23.9% and 23.5% in males and 24.4% and 24.6% in females, respectively), compared to other Hispanic lung cancer groups. Foreign-born Mexicans have the lowest proportions of private insurance (17.3% for males and 16.7% for females), compared to other Hispanic lung cancer groups. Lung cancer patients with microscopic verification were lowest among Hispanic females at 90.1%, comprising detailed groups with proportions varying from 83.3% for foreign-born Mexican females and 91.5% for Puerto Rican females.

The distribution of lung cancer cases by SEER stage at diagnosis show higher proportions of late-stage lung cancer among Black and API populations (52.9% and 57.4% for males and 46.8% and 58.5% for females, respectively); however, a larger distant stage percentage for APIs is partially explained by the lower proportion of unknown stage. Among Hispanic lung cancer patients, higher proportions of distant stage at diagnosis were observed among Mexicans (55.8% for males and 61.0% for females) and Central Americans (55.8% and 52.1%).

Histologically, lung cancer cases show larger proportions of adenocarcinomas to be among APIs (56.5% and 68.2% for males and females, respectively), Caribbean-born Blacks (50.0% and 58.8%), Dominicans (49.9% and 58.0%), Central Americans (53.2% and 58.9%), and US-born Mexican women (58.1%). Among Black females with lung cancer, those who are US-born have larger proportions for squamous cell carcinoma (18.5%) and small cell carcinoma (10.2%) compared to those who are Caribbean-born (8.8% and 3.0%, respectively).

Significant disparities across lung cancer cases by racial–ethnic group in census tract poverty level show the highest overall proportions of high poverty for Blacks (58.0% for both males and females). Proportions of high poverty among Blacks by detailed racial–ethnic group reveal US-born Black males and females at 59.6% and 57.4%, respectively, and Caribbean-born Black males at 47.1% and females at 37.6%. The lowest proportions of high poverty among lung cancer cases are for API (20.3% for both males and females), White (21.7% for both males and females), and South American (23.3% for males and 23.0% for females) populations.

Overall, age-adjusted lung cancer incidence rates (AAIR) per 100,000 people (Table 3) were highest among White males and females (71.4 and 61.0, respectively) and Black males (63.5), intermediate for Black females (36.0) and the Hispanic population (49.8 and 26.4 for males and females, respectively), and lowest among the API population (25.5 and 18.8, for males and females). Incidence rates per 100,000 people by racial–ethnic group in Florida show that lung cancer rates for US-born Blacks (AAIR: 81.7; 95%CI: 79.3–84.2 and AAIR: 46.4; 95%CI: 44.8–48.0, for males and females) are over three times that of Caribbean-born Blacks (AAIR: 26.2; 95%CI: 24.1–28.4 and AAIR: 14.9; 95%CI: 13.6–16.3, for males and females) for both sexes; that Cuban males (AAIR: 65.6; 95%CI: 63.6–67.6) have over double the incidence of Central and South American males (AAIR: 30.8; 95%CI: 28.8–32.9); and that lung cancer rates in US-born Mexicans in Florida (AAIR: 59.8; 95%CI: 49.5–71.3 and AAIR: 23.5; 95%CI: 18.5–29.2, for males and females) are approximately double that of foreign-born Mexicans (AAIR: 22.3; 95%CI: 17.4–28.2 and AAIR: 12.2; 95%CI: 9.1–16.0, respectively), for both sexes. Incidence rate ratios, using the White population as reference, are significantly lower for all major racial–ethnic population groups (Black, API, and Hispanic). However, detailed IRRs (Table 3) reveal US-born Black men with lung cancer rates 14% higher than White men (IRR: 1.14; 95%CI: 1.11–1.18) and over three times that of those who are Caribbean-born (IRR: 3.12; 95%CI: 2.80–3.40). Among Hispanic groups, Cuban males have the highest rate of lung cancer (AAIR: 65.6; 95%CI: 63.6–67.6), 2.72 (IRR 95%CI: 2.31–3.19) times higher than Central American males. Foreign-born Mexican females (IRR: 0.20; 95%CI: 0.15–0.26) had 80% lower rates of lung cancer, compared to White females. Detailed age-adjusted incidence rates for microscopically verified, adenocarcinoma, and late-stage disease can be found in Appendix A.

## 4. Discussion

This novel analysis of lung cancer incidence rates during 2012–2018 in Florida by detailed race–ethnicity provides significant insight into this important disease in US Hispanic and Black minority populations. In these thus far unstudied detailed populations, we demonstrate remarkable variation in lung cancer risk within each racial–ethnic group, most apparently Cubans versus Central Americans, US-born Black persons versus Caribbean-born Black persons, and US-born Mexicans versus foreign-born Mexicans. Such marked heterogeneity is important, as it supports recommendations to limit consideration of race–ethnicity as strictly blocked groups (White, Black, API, Hispanic), in an effort to reduce inequities in scientific literature [23]. In jurisdictions where populations are continually diversifying, like the US, it is critical to consider enhanced granularity in racial–ethnic stratification to understand population-level lung cancer disparities and apply such knowledge to strengthen public health programming targeted to marginalized detailed populations often hidden by large racial–ethnic grouping.

Our analyses show higher overall lung cancer rates in males than females, which reflects secular trends in the prevalence of smoking by sex [13,24]. Moreover, lung cancer incidence rates for Florida Hispanics are significantly higher than the national rates for US Hispanics during the same period, as per the Centers for Disease Control and Prevention WONDER website (AAIR: 37.7 per 100,000 for males and AAIR: 25.1 per 100,000 for females) [2]. This is largely due to a higher lung cancer rate in Cubans (AAIR: 65.6; 95%CI: 63.6–67.6 for males and AAIR: 31.7; 95%CI: 30.5–33.0 for females), the largest Hispanic group in Florida [21], as found in this study. Differences in lung cancer risk among specific racial–ethnic groups are primarily associated with smoking habits and support the potential need for smoking cessation programs tailored specifically to such populations as Cubans and US-born Blacks, populations that have been historical targets of tobacco industry marketing [25,26] and have significant barriers to tobacco counseling and cessation treatments [27,28,29]. Focusing interventions on such groups is also important for appropriate resource allocation and budgeting to help mitigate monies or efforts from being spread too thin among large umbrella populations.

Although lung cancer incidence rates are overwhelmingly driven by prevalence of current or past smoking, it is important to consider that lung cancer also occurs among never-smokers. The characterization of lung cancer among never-smokers displays distinct patterns with higher proportions of adenocarcinomas, and for lung cancer patients with no smoking history, there is the potential for improved prognosis due to the availability of targeted therapies [30]. In a previous study [31], the proportions of never-smokers, in relation to the total lung cancer patients, varied remarkably by sex and race–ethnicity. High proportions of never-smokers among lung cancer cases were found for APIs, Caribbean-born Blacks, foreign-born Mexicans, Dominicans, and Central Americans. In contrast, lower proportions of never-smokers among lung cancer cases were among US-born Blacks, US-born Mexicans, Cubans, and Puerto Ricans, as well as Whites. The previous study [31] suggests that never-smoker proportions among lung cancer cases are correlated with the smoking prevalence within each detailed population. Correspondingly, in our study, we found higher proportions of adenocarcinoma for APIs, Caribbean-born Blacks, Dominicans, and Central Americans. This is likely because adenocarcinomas are more common among never-smokers [30,31], whereas among current or past smoker populations, in which lung cancer occurs more often, squamous cell and small cell carcinomas dominate [31]. Adenocarcinoma risk has also been found to be higher among females [30] and in people exposed to environmental pollution [14]. Considering that, for some of these detailed racial–ethnic groups, lung cancer cases occur disproportionately among never-smokers [31], further research among never-smokers by detailed racial–ethnicity is necessary, especially among females and accounting for other environmental factors involved in lung cancer, such as asbestos, radon, environmental tobacco smoke, and air pollution [32,33].

The US is becoming increasingly diverse, due largely to immigration and lower birth rates, with US youth and children at the leading edge of the nation’s growing heterogeneity as the population of racial–ethnic minority children increased by 11.8% over the last decade to 38.5 million in 2020, when 52.7% of U.S. population under age 18 belonged to a minority group [34,35]. This emerging US diversity demands an even larger focus on primordial lung cancer prevention efforts tailored to racial–ethnic minorities, particularly Black and Hispanic youth and children. Previous US studies have shown that acculturation of foreign-born populations (i.e., first-generation) is associated with higher prevalence of smoking over time [36,37,38], considerably increasing lung cancer risk. This relationship is even higher among second- and third-generation individuals (i.e., US-born) [36,37,38] and extends to e-cigarette usage (i.e., vaping) [39], which is gaining in popularity, especially among US youth. While not (yet) linked to an increased risk of lung cancer, e-cigarette usage has been shown to lead to transference to and increases in combustible tobacco use, reversing a decline observed in recent years [40]. Considering the increased diversity of those at risk of future lung cancer and the uncertain impacts of e-cigarette usage, it is important to monitor lung cancer trends in detailed racial–ethnic groups that will likely be affected. Likewise, public health efforts are needed to curtail initiation of smoking in distinct younger populations, perhaps with culturally sensitive and socially conscious anti-tobacco programming.

When aggregated, the Florida Black population shows a lower risk of lung cancer, compared to national rates for Blacks [2], exposing a previously identified phenomenon called the ‘healthy immigrant effect’ that masks significant racial–ethnic disparities by nativity [8,9,41]. Lower rates of lung cancer in foreign-born Black populations, combined with the substantial proportion of foreign-born Black persons in Florida, produce a lower, averaged rate for aggregate Blacks, compared to national rates [2,41]. This effect is also evident in the Florida Mexican population, where lower rates of lung cancer in those who are foreign-born offset those among US-born Mexicans, aligning with previous literature [6,42]. Revealing these differences in Mexican groups by nativity is important, due to the vast weight that this population carries nationally, although not necessarily in Florida. Marked differences between US-born and foreign-born populations highlight the importance of recognizing such health ‘paradoxes’ among immigrants when calculating cancer incidence rates for the aggregate racial–ethnic groups to which they belong. To better understand US health disparities, it is important to consider not only detailed racial–ethnic grouping, but also place of birth, if possible, during scientific research, to account for possible differences between specific populations.

Within Hispanics, we found a higher proportions of late-stage lung cancer diagnoses among Mexican and Central American populations. Such notable differences in the distribution of stage at diagnosis are highly suggestive of disparities within Hispanics that are critical for understanding cancer patterns in this population. US immigrants from Mexico and Central America tend to be highly vulnerable populations, with lower socio-economic status and lower levels of education and English language proficiency compared to other foreign-born groups, all considerably lower than their US-born counterparts [43]. Tailored public health measures, such as lung cancer screening and prevention strategies, are needed for more vulnerable groups.

A thorough literature review returns few published studies evaluating lung cancer incidence across detailed race–ethnicity, and existing studies primarily focus on non-Hispanic API racial groups [44,45], excluding non-Hispanic Black and Hispanic groups. Previous AAIR calculations for Florida Cubans, Mexicans, and Puerto Ricans during 1999–2001 [6] show higher rates of lung cancer for specific Hispanic groups (i.e., Puerto Ricans, Mexicans, and Cuban males), aligning with overall declines in lung cancer compared to our newly determined rates (2012–2018). Stable lung cancer rates among Cuban females were the exception during the two periods, which emphasizes the value of within-Hispanic group analyses and warrants further exploration.

The primary strength of this study is its utilization of all-inclusive, population-based data in the calculation of lung cancer incidence rates and rate ratios which, compared to a cohort study, minimizes error and produces more accurate comparisons across populations, given selection factors in current cohorts [46,47]. Significant diversity across racial–ethnic groups is more accurately characterized through utilization of the entire Florida lung cancer case population, as compared to differences between proportions based on samples of problematic representativeness, which are limited indicators as they do not account for the population-at-risk as do rates. Notwithstanding, we do consider the study limitations. This analysis fails to incorporate population rates for ever-smokers and never-smokers due to lack of data on smoking by detailed race–ethnicity in the general population in the state of Florida. Information was also unavailable for other known lung cancer risk factors and occupational hazards (e.g., asbestos, radon, environmental tobacco smoke, air pollution). Furthermore, a portion of the data on specific racial–ethnic groups was derived from multiple imputation due to missing values. Lastly, considering that the populations being compared have diverse age structures, using the US standard population for all groups could have affected age adjustments.

## 5. Conclusions

The disparities in lung cancer incidence in Hispanic and non-Hispanic Black populations, presented here, highlight the heterogeneity of lung cancer in rapidly expanding, racial–ethnic populations that is often hidden by aggregation of groups. AAIRs among Blacks and Hispanics vary approximately 3-fold in their more detailed groups, as opposed to the monotonous aggregate groups, requiring more refined approaches in the fight against lung cancer. The implications of these findings are significant for public health surveillance and for clinical professionals working with diverse US populations. The characterization of disparate lung cancer rates between groups helps to identify local populations that may benefit from targeted prevention programming and education and from enhanced lung cancer screening. Areas for future research include the calculation of specific rates of lung cancer in never-smokers by detailed racial–ethnic population, especially among females; further research into the effects of environmental factors on lung cancer risk in diverse populations; and the replication of this study in other states with varied racial–ethnic populations.

## Figures and Tables

**Table 1 cancers-15-02164-t001:** Demographic and Clinical Characteristics of Lung Cancer Cases by Race–Ethnicity and Non-Hispanic Black Detailed Race–Ethnicity and Sex. Florida, 2012–2018.

	Non-Hispanic White	Non-Hispanic Black ^a^	*Non-Hispanic Black US-Born*	*Non-Hispanic Black Caribbean-Born*	Non-Hispanic API	Hispanic	All Race–Ethnicities ^a^
**Male and Female Combined (N)**	96,131	9710	8452	1169	1030	12,062	120,550
**MALE**							
**Total (n (%))**	49,043 (51.0%)	5553 (57.2%)	4852 (57.4%)	654 (55.9%)	512 (49.7%)	7038 (58.3%)	63,143 (52.4%)
**Median Age (Years (IQR))**	71 (14)	67 (15)	67 (15)	68 (16)	68 (16)	70 (15)	70 (15)
**Age Group (n (%))**							(*p* < 0.001) ^b^
15–44	318 (0.7%)	107 (1.9%)	84 (1.7%)	18 (2.7%)	10 (2.0%)	99 (1.4%)	547 (0.9%)
45–54	2713 (5.5%)	521 (9.4%)	458 (9.5%)	57 (8.7%)	51 (10.0%)	472 (6.7%)	3830 (6.1%)
55–64	10,002 (20.4%)	1729 (31.1%)	1553 (32.0%)	160 (24.4%)	121 (23.6%)	1459 (20.7%)	13,537 (21.4%)
65–74	17,392 (35.5%)	1843 (33.2%)	1613 (33.3%)	218 (33.4%)	187 (36.5%)	2466 (35.0%)	22,277 (35.3%)
75–84	13,982 (28.5%)	1068 (19.2%)	909 (18.7%)	155 (23.7%)	108 (21.1%)	1968 (28.0%)	17,351 (27.5%)
85+	4636 (9.5%)	285 (5.1%)	235 (4.8%)	47 (7.1%)	35 (6.8%)	574 (8.2%)	5601 (8.9%)
**Histology (n (%))**							(*p* < 0.001) ^b^
Adenocarcinoma	18,595 (37.9%)	2251 (40.5%)	1901 (39.2%)	327 (50.0%)	289 (56.5%)	2921 (41.5%)	24,298 (38.5%)
Squamous Cell Carcinoma	12,501 (25.5%)	1319 (23.8%)	1198 (24.7%)	112 (17.2%)	89 (17.4%)	1438 (20.4%)	15,486 (24.5%)
Non-Small Cell Lung Cancer NOS	2596 (5.3%)	357 (6.4%)	329 (6.8%)	24 (3.7%)	30 (5.9%)	442 (6.3%)	3462 (5.5%)
Small Cell Carcinoma	5569 (11.4%)	503 (9.1%)	456 (9.4%)	44 (6.7%)	41 (8.0%)	690 (9.8%)	6859 (10.9%)
Large Cell Carcinoma	683 (1.4%)	105 (1.9%)	99 (2.0%)	6 (0.9%)	† (0.6%)	85 (1.2%)	882 (1.4%)
Other Specified	2253 (4.6%)	248 (4.5%)	195 (4.0%)	50 (7.6%)	19 (3.7%)	450 (6.4%)	3007 (4.8%)
Unspecified	6846 (14.0%)	770 (13.9%)	674 (13.9%)	92 (14.0%)	41 (8.0%)	1012 (14.4%)	9149 (14.5%)
**Microscopically Verified (n (%))**							(*p* < 0.001) ^b^
Yes	44,213 (90.2%)	5064 (91.2%)	4418 (91.1%)	602 (92.0%)	493 (96.3%)	6409 (91.1%)	56,735 (89.9%)
No	4830 (9.8%)	489 (8.8%)	434 (8.9%)	52 (8.0%)	19 (3.7%)	629 (8.9%)	6408 (10.1%)
**SEER Stage at Diagnosis (n (%))**							(*p* < 0.001) ^b^
Localized	10,196 (20.8%)	859 (15.5%)	750 (15.5%)	100 (15.3%)	88 (17.2%)	1276 (18.1%)	12,562 (19.9%)
Regional	11,061 (22.6%)	1191 (21.5%)	1052 (21.7%)	133 (20.3%)	103 (20.1%)	1587 (22.6%)	14,062 (22.3%)
Distant	22,779 (46.5%)	2937 (52.9%)	2554 (52.6%)	354 (54.1%)	294 (57.4%)	3360 (47.7%)	29,631 (46.9%)
Unknown	5007 (10.2%)	566 (10.2%)	496 (10.2%)	68 (10.4%)	27 (5.3%)	815 (11.6%)	6888 (10.9%)
**Census Tract Poverty Level (n (%)) ^c^**							(*p* < 0.001) ^b^
High	10,633 (21.7%)	3218 (58.0%)	2890 (59.6%)	308 (47.1%)	104 (20.3%)	2738 (38.9%)	16,936 (26.8%)
Medium	19,413 (39.6%)	1552 (28.0%)	1306 (26.9%)	229 (35.0%)	205 (40.0%)	2637 (37.5%)	24,189 (38.3%)
Low	5101 (10.4%)	174 (3.1%)	487 (10.0%)	86 (13.1%)	77 (15.0%)	1275 (18.1%)	5804 (9.2%)
Very Low	13,650 (27.8%)	581 (10.5%)	142 (2.9%)	31 (4.7%)	125 (24.4%)	367 (5.2%)	15,912 (25.2%)
Unknown	246 (0.5%)	28 (0.5%)	28 (0.6%)	† (0.1%)	† (0.2%)	21 (0.3%)	302 (0.5%)
**Insurance (n (%))**							(*p* < 0.001) ^b^
Private	14,069 (28.7%)	1189 (21.4%)	1037 (21.4%)	138 (21.1%)	170 (33.2%)	1731 (24.6%)	17,292 (27.4%)
Medicare	25,007 (51.0%)	2287 (41.2%)	2015 (41.5%)	261 (39.8%)	190 (37.1%)	2956 (42.0%)	30,734 (48.7%)
Medicaid	3924 (8.0%)	1157 (20.8%)	1030 (21.2%)	120 (18.3%)	92 (18.0%)	1268 (18.0%)	6508 (10.3%)
No insurance	1406 (2.9%)	351 (6.3%)	270 (5.6%)	79 (12.0%)	33 (6.5%)	394 (5.6%)	2207 (3.5%)
Unknown	4637 (9.5%)	569 (10.3%)	501 (10.3%)	57 (8.8%)	27 (5.3%)	689 (9.8%)	6402 (10.1%)
**FEMALE ^d^**							
**Total (n (%))**	47,088 (49.0%)	4157 (42.8%)	3600 (42.6%)	515 (44.1%)	518 (50.3%)	5024 (41.7%)	57,407 (47.6%)
**Median Age (Years (IQR))**	71 (14)	67 (19)	67 (17)	69 (17)	67 (16)	70 (16)	70 (15)
**Age Group (n (%))**							(*p* < 0.001) ^b^
15–44	385 (0.8%)	103 (2.5%)	74 (2.1%)	20 (3.8%)	20 (3.9%)	106 (2.1%)	624 (1.1%)
45–54	2794 (5.9%)	420 (10.1%)	364 (10.1%)	48 (9.3%)	51 (9.9%)	406 (8.1%)	3709 (6.5%)
55–64	9294 (19.7%)	1186 (28.5%)	1076 (29.9%)	101 (19.6%)	114 (22.0%)	1047 (20.8%)	11,771 (20.5%)
65–74	16,622 (35.3%)	1246 (30.0%)	1073 (29.8%)	166 (32.2%)	182 (35.1%)	1650 (32.8%)	19,928 (34.7%)
75–84	13,110 (27.8%)	874 (21.0%)	740 (20.6%)	128 (24.8%)	126 (24.3%)	1337 (26.6%)	15,601 (27.2%)
85+	4883 (10.4%)	328 (7.9%)	272 (7.6%)	53 (10.3%)	25 (4.8%)	478 (9.5%)	5774 (10.1%)
**Histology (n (%))**							(*p* < 0.001) ^b^
Adenocarcinoma	21,571 (45.8%)	2028 (48.8%)	1703 (47.3%)	303 (58.8%)	353 (68.2%)	2514 (50.0%)	26,661 (46.4%)
Squamous Cell Carcinoma	8096 (17.2%)	715 (17.2%)	667 (18.5%)	46 (8.8%)	57 (11%)	616 (12.3%)	9547 (16.6%)
Non-Small Cell Lung Cancer NOS	2056 (4.4%)	224 (5.4%)	203 (5.6%)	19 (3.6%)	21 (4.1%)	223 (4.4%)	2544 (4.4%)
Small Cell Carcinoma	6215 (13.2%)	382 (9.2%)	367 (10.2%)	15 (3.0%)	26 (5.0%)	448 (8.9%)	7112 (12.4%)
Large Cell Carcinoma	550 (1.2%)	56 (1.4%)	49 (1.4%)	† (1.1%)	† (0.6%)	54 (1.1%)	669 (1.2%)
Other Specified	2745 (5.8%)	263 (6.3%)	199 (5.5%)	57 (11.0%)	28 (5.4%)	432 (8.6%)	3508 (6.1%)
Unspecified	5855 (12.4%)	489 (11.8%)	413 (11.5%)	71 (13.7%)	30 (5.8%)	737 (14.7%)	7366 (12.8%)
**Microscopically Verified (n (%))**							(*p* < 0.001) ^b^
Yes	42,971 (91.3%)	3847 (92.5%)	3343 (92.9%)	467 (90.7%)	501 (96.7%)	4528 (90.1%)	52,236 (91.0%)
No	4117 (8.7%)	310 (7.5%)	257 (7.1%)	48 (9.3%)	17 (3.3%)	496 (9.9%)	5171 (9.0%)
**SEER Stage at Diagnosis (n (%))**							(*p* < 0.001) ^b^
Localized	12,161 (25.8%)	933 (22.4%)	817 (22.7%)	108 (21%)	112 (21.6%)	1176 (23.4%)	14,490 (25.2%)
Regional	10,564 (22.4%)	924 (22.2%)	822 (22.8%)	96 (18.7%)	90 (17.4%)	1066 (21.2%)	12,723 (22.2%)
Distant	20,422 (43.4%)	1945 (46.8%)	1675 (46.5%)	249 (48.3%)	303 (58.5%)	2240 (44.6%)	25,062 (43.7%)
Unknown	3941 (8.4%)	355 (8.5%)	286 (7.9%)	62 (12.1%)	13 (2.5%)	542 (10.8%)	5132 (8.9%)
**Census Tract Poverty Level (n (%)) ^c^**							(*p* < 0.001) ^b^
High	9306 (21.7%)	2277 (58.0%)	2065 (57.4%)	194 (37.6%)	108 (20.3%)	1736 (34.6%)	13,552 (23.6%)
Medium	18,795 (39.6%)	1202 (28.0%)	991 (27.5%)	201 (39.0%)	186 (40.0%)	1945 (38.7%)	22,358 (38.9%)
Low	13,654 (27.8%)	512 (10.5%)	416 (11.6%)	87 (16.8%)	149 (24.4%)	981 (19.5%)	15,491 (27%)
Very Low	5142 (10.4%)	150 (3.1%)	113 (3.1%)	33 (6.4%)	73 (15.0%)	346 (6.9%)	5778 (10.1%)
Unknown	191 (0.5%)	16 (0.5%)	15 (0.4%)	1 (0.2%)	2 (0.2%)	16 (0.3%)	228 (0.4%)
Insurance (n (%))							(*p* < 0.001) ^b^
Private	14,674 (31.2%)	1095 (26.3%)	932 (25.9%)	146 (28.3%)	160 (30.9%)	1348 (26.8%)	17,388 (30.3%)
Medicare	23,247 (49.4%)	1613 (38.8%)	1432 (39.8%)	177 (34.4%)	251 (48.5%)	2002 (39.9%)	27,268 (47.5%)
Medicaid	4136 (8.8%)	899 (21.6%)	800 (22.2%)	90 (17.6%)	53 (10.2%)	904 (18.0%)	6029 (10.5%)
No insurance	1107 (2.4%)	204 (4.9%)	151 (4.2%)	47 (9.0%)	27 (5.2%)	273 (5.4%)	1626 (2.8%)
Unknown	3924 (8.3%)	346 (8.3%)	286 (7.9%)	55 (10.7%)	27 (5.2%)	497 (9.9%)	5096 (8.9%)

^a^. Includes all cases included here and other race–ethnicity; ^b^. *p*-value obtained from chi-square test for differences between major racial–ethnic groups only (i.e., non-Hispanic White, Black, and API and Hispanic); ^c^. Census tract poverty level is defined as the proportion of the population living below: very low (0% to < 5%), low (5% to < 10%), medium (10% to < 20%), high (20% to < 100%); ^d^. Includes n = 41 identified as non-binary. † Not reported due to fewer than 10 cases in group. Abbreviation: API: Asian/Pacific Islander; N: number; IQR: interquartile range; NOS: not otherwise specified; SEER: Surveillance, Epidemiology, and End Results Program.

**Table 2 cancers-15-02164-t002:** Demographic and Clinical Characteristics of Lung Cancer Cases by Hispanic Detailed Race–Ethnicity and Sex. Florida, 2012–2018.

	Mexican ^a^	*Mexican* *US-Born*	*Mexican* *Foreign-Born*	Puerto Rican	Cuban	Dominican	Central American	South American	All Hispanic ^a^
**Male and Female Combined (N)**	429	265	164	2250	6680	405	466	1567	12,062
**MALE**									
**Total (n (%))**	276 (64.3%)	172 (64.9%)	104 (63.4%)	1224 (54.4%)	4189 (62.7%)	207 (51.1%)	208 (44.6%)	778 (49.6%)	7038 (58.3%)
**Median Age (Years (IQR))**	67 (19)	67 (20)	66 (18)	69 (15)	71 (14)	69 (14)	68 (16)	69 (14)	70 (15)
**Age Group (n (%))**									(*p* < 0.001) ^b^
15–44	† (2.6%)	† (1.7%)	† (4.8%)	26 (2.1%)	39 (0.9%)	† (2.4%)	† (3.1%)	10 (1.3%)	99 (1.4%)
45–54	35 (12.6%)	21 (12.2%)	13 (12.5%)	90 (7.3%)	238 (5.7%)	14 (6.8%)	21 (10.2%)	65 (8.3%)	472 (6.7%)
55–64	73 (26.7%)	44 (25.6%)	29 (27.9%)	275 (22.5%)	819 (19.5%)	45 (22%)	47 (22.5%)	171 (22.0%)	1459 (20.7%)
65–74	75 (27.1%)	45 (26.2%)	30 (28.8%)	425 (34.7%)	1476 (35.2%)	69 (33.6%)	63 (30.2%)	309 (39.7%)	2466 (35.0%)
75–84	62 (22.5%)	43 (25.0%)	19 (18.3%)	325 (26.5%)	1240 (29.6%)	55 (26.8%)	55 (26.3%)	185 (23.8%)	1968 (28.0%)
85+	24 (8.6%)	16 (9.3%)	† (7.7%)	84 (6.8%)	378 (9.0%)	18 (8.5%)	16 (7.7%)	39 (5.0%)	574 (8.2%)
**Histology (n (%))**									(*p* < 0.001) ^b^
Adenocarcinoma	110 (40.0%)	67 (39.0%)	43 (41.3%)	506 (41.3%)	1615 (38.5%)	103 (49.9%)	414 (53.2%)	93 (44.8%)	2921 (41.5%)
Squamous Cell Carcinoma	67 (24.3%)	38 (22.1%)	29 (27.9%)	278 (22.7%)	912 (21.8%)	35 (16.9%)	91 (11.7%)	28 (13.5%)	1438 (20.4%)
Non-Small Cell Lung Cancer NOS	17 (6.0%)	12 (7.0%)	† (4.8%)	82 (6.7%)	261 (6.2%)	14 (6.7%)	43 (5.6%)	19 (9.0%)	442 (6.3%)
Small Cell Carcinoma	28 (10.1%)	19 (11.0%)	† (8.7%)	125 (10.2%)	418 (10.0%)	18 (8.8%)	60 (7.7%)	20 (9.6%)	690 (9.8%)
Large Cell Carcinoma	† (0.5%)	† (1.2%)	† (0.0%)	10 (0.8%)	53 (1.3%)	† (1.2%)	15 (1.9%)	† (1.5%)	85 (1.2%)
Other Specified	14 (5.2%)	† (3.5%)	† (7.7%)	68 (5.6%)	281 (6.7%)	† (3.4%)	55 (7.1%)	15 (7.4%)	450 (6.4%)
Unspecified	38 (13.8%)	28 (16.3%)	10 (9.6%)	156 (12.7%)	649 (15.5%)	27 (13.2%)	100 (12.8%)	30 (14.2%)	1012 (14.4%)
**Microscopically Verified (n (%))**									(*p* = 0.575) ^b^
Yes	250 (90.6%)	153 (89.0%)	97 (93.3%)	1122 (91.7%)	3791 (90.5%)	187 (90.3%)	191 (91.8%)	718 (92.3%)	6409 (91.1%)
No	26 (9.4%)	19 (11.0%)	7 (6.7%)	102 (8.3%)	398 (9.5%)	20 (9.7%)	17 (8.2%)	60 (7.7%)	629 (8.9%)
**SEER Stage at Diagnosis (n (%))**									(*p* < 0.001) ^b^
Localized	30 (10.8%)	20 (11.6%)	10 (9.6%)	210 (17.2%)	788 (18.8%)	42 (20.3%)	36 (17.2%)	124 (16.0%)	1276 (18.1%)
Regional	65 (23.5%)	45 (26.2%)	20 (19.2%)	278 (22.7%)	966 (23.1%)	44 (21.4%)	35 (16.7%)	159 (20.5%)	1587 (22.6%)
Distant	154 (55.8%)	84 (48.8%)	70 (67.3%)	608 (49.7%)	1923 (45.9%)	95 (46.0%)	116 (55.8%)	411 (52.8%)	3360 (47.7%)
Unknown	27 (9.9%)	23 (13.4%)	† (3.8%)	128 (10.5%)	512 (12.2%)	25 (12.3%)	21 (10.3%)	84 (10.7%)	815 (11.6%)
**Census Tract Poverty Level (n (%)) ^c^**									(*p* < 0.001) ^b^
High	107 (38.7%)	60 (34.9%)	47 (45.2%)	412 (33.6%)	1853 (44.2%)	73 (35.2%)	78 (37.4%)	181 (23.3%)	2738 (38.9%)
Medium	97 (35.1%)	59 (34.3%)	38 (36.5%)	457 (37.4%)	1526 (36.4%)	84 (40.7%)	84 (40.4%)	317 (40.7%)	2637 (37.5%)
Low	54 (19.6%)	39 (22.7%)	15 (14.4%)	257 (21.0%)	643 (15.4%)	36 (17.4%)	36 (17.3%)	212 (27.2%)	1275 (18.1%)
Very Low	16 (6.0%)	13 (7.6%)	† (2.9%)	91 (7.5%)	157 (3.8%)	14 (6.7%)	† (4.3%)	67 (8.6%)	367 (5.2%)
Unknown	† (0.7%)	† (0.6%)	† (1.0%)	† (0.6%)	† (0.2%)	† (0.1%)	† (0.5%)	† (0.2%)	21 (0.3%)
**Insurance (n (%))**									(*p* < 0.001) ^b^
Private	66 (23.9%)	48 (27.9%)	18 (17.3%)	310 (25.3%)	985 (23.5%)	51 (24.5%)	52 (24.8%)	217 (27.9%)	1731 (24.6%)
Medicare	89 (32.2%)	62 (36.0%)	28 (26.9%)	565 (46.1%)	1769 (42.2%)	88 (42.8%)	70 (33.8%)	304 (39.0%)	2956 (42.0%)
Medicaid	60 (21.9%)	32 (18.6%)	28 (26.9%)	179 (14.6%)	836 (20.0%)	39 (19.1%)	34 (16.2%)	105 (13.5%)	1268 (18.0%)
No insurance	27 (9.9%)	† (4.7%)	19 (18.3%)	48 (3.9%)	203 (4.8%)	11 (5.5%)	25 (11.9%)	74 (9.5%)	394 (5.6%)
Unknown	34 (12.2%)	22 (12.8%)	11 (10.6%)	123 (10.0%)	396 (9.4%)	17 (8.1%)	28 (13.3%)	78 (10.1%)	689 (9.8%)
**FEMALE ^d^**									
**Total (n)**	153 (35.7%)	93 (35.1%)	60 (36.6%)	1026 (45.6%)	2491 (37.3%)	198 (48.9%)	258 (55.4%)	789 (50.4%)	5024 (41.7%)
**Median Age (Years (IQR))**	68 (19)	69 (17)	68 (19)	69 (17)	70 (15)	69 (15)	67 (19)	69 (17)	70 (16)
**Age Group (n (%))**									(*p* < 0.001) ^b^
15–44	† (4.4%)	† (4.3%)	† (5.0%)	35 (3.4%)	30 (1.2%)	† (1.9%)	10 (3.9%)	18 (2.3%)	106 (2.1%)
45–54	14 (9.0%)	† (9.7%)	† (8.3%)	89 (8.6%)	183 (7.4%)	14 (7.1%)	32 (12.6%)	66 (8.4%)	406 (8.1%)
55–64	33 (21.8%)	17 (18.3%)	16 (26.7%)	217 (21.2%)	500 (20.1%)	41 (20.9%)	66 (25.6%)	168 (21.3%)	1047 (20.8%)
65–74	50 (32.4%)	34 (36.6%)	16 (26.7%)	301 (29.3%)	867 (34.8%)	67 (34%)	66 (25.7%)	257 (32.6%)	1650 (32.8%)
75–84	33 (21.7%)	18 (19.4%)	15 (25.0%)	292 (28.4%)	666 (26.8%)	57 (28.9%)	60 (23.3%)	201 (25.5%)	1337 (26.6%)
85+	16 (10.7%)	11 (11.8%)	† (8.3%)	93 (9.0%)	245 (9.9%)	14 (7.3%)	23 (9.1%)	77 (9.8%)	478 (9.5%)
**Histology (n (%))**									(*p* < 0.001) ^b^
Adenocarcinoma	82 (53.3%)	54 (58.1%)	28 (46.7%)	473 (46.1%)	1187 (47.6%)	115 (58.0%)	465 (58.9%)	127 (49.3%)	2514 (50.0%)
Squamous Cell Carcinoma	18 (12.0%)	† (9.7%)	† (12.0%)	164 (16.0%)	310 (12.4%)	12 (6.0%)	67 (8.5%)	29 (11.3%)	616 (12.3%)
Non-Small Cell Lung Cancer NOS	† (3.9%)	† (3.2%)	† (5.0%)	57 (5.5%)	108 (4.3%)	† (3.0%)	24 (3.1%)	21 (8%)	223 (4.4%)
Small Cell Carcinoma	16 (10.3%)	11 (11.8%)	† (8.3%)	101 (9.9%)	240 (9.7%)	16 (8.1%)	48 (6.1%)	17 (6.7%)	448 (8.9%)
Large Cell Carcinoma	† (0.7%)	† (1.1%)	† (0.0%)	† (0.6%)	34 (1.4%)	† (2.0%)	† (1.0%)	† (0.5%)	54 (1.1%)
Other Specified	† (3.8%)	† (3.2%)	† (5.0%)	91 (8.8%)	226 (9.1%)	14 (7.0%)	62 (7.9%)	23 (9.0%)	432 (8.6%)
Unspecified	24 (16.1%)	12 (12.9%)	12 (20.0%)	134 (13.1%)	387 (15.5%)	31 (15.9%)	114 (14.5%)	40 (15.4%)	737 (14.7%)
**Microscopically Verified (n (%))**									(*p* < 0.001) ^b^
Yes	133 (86.9%)	83 (89.2%)	50 (83.3%)	939 (91.5%)	2226 (89.4%)	177 (89.4%)	233 (90.3%)	714 (90.5%)	4528 (90.1%)
No	20 (13.1%)	10 (10.8%)	10 (16.7%)	87 (8.4%)	265 (10.6%)	21 (10.6%)	25 (9.7%)	75 (9.5%)	496 (9.9%)
**SEER Stage at Diagnosis (N (%))**									(*p* < 0.001) ^b^
Localized	22 (14.6%)	15 (16.1%)	† (11.7%)	239 (23.3%)	587 (23.6%)	56 (28.2%)	50 (19.5%)	181 (23%)	1176 (23.4%)
Regional	23 (15.0%)	16 (17.2%)	† (11.7%)	219 (21.3%)	540 (21.7%)	42 (21.3%)	49 (18.9%)	167 (21.1%)	1066 (21.2%)
Distant	94 (61.0%)	55 (59.1%)	39 (65.0%)	472 (46.0%)	1066 (42.8%)	80 (40.6%)	134 (52.1%)	361 (45.8%)	2240 (44.6%)
Unknown	14 (9.4%)	† (7.5%)	† (11.7%)	97 (9.5%)	298 (12.0%)	20 (9.9%)	24 (9.5%)	80 (10.1%)	542 (10.8%)
**Census Tract Poverty Level (n (%))** ^c^									(*p* < 0.001) ^b^
High	59 (38.5%)	33 (35.5%)	26 (43.3%)	322 (31.4%)	994 (39.9%)	67 (33.6%)	92 (35.5%)	181 (23.0%)	1736 (34.6%)
Medium	49 (31.8%)	36 (38.7%)	13 (21.7%)	429 (41.8%)	928 (37.3%)	80 (40.5%)	92 (35.7%)	316 (40.1%)	1945 (38.7%)
Low	36 (23.3%)	20 (21.5%)	16 (26.7%)	196 (19.1%)	440 (17.7%)	38 (19.2%)	48 (18.6%)	196 (24.9%)	981 (19.5%)
Very Low	9 (6.3%)	† (4.3%)	† (8.3%)	75 (7.3%)	122 (4.9%)	13 (6.5%)	24 (9.4%)	93 (11.8%)	346 (6.9%)
Unknown	† (0.0%)	† (0.0%)	† (0.0%)	† (0.4%)	† (0.3%)	† (0.1%)	† (0.8%)	† (0.3%)	16 (0.3%)
**Insurance (n (%))**									(*p* < 0.001) ^b^
Private	37 (24.4%)	26 (28.0%)	10 (16.7%)	298 (29.0%)	613 (24.6%)	55 (27.6%)	80 (31.2%)	231 (29.3%)	1348 (26.8%)
Medicare	51 (33.3%)	34 (36.6%)	16 (26.7%)	396 (38.6%)	1060 (42.6%)	84 (42.2%)	92 (35.6%)	270 (34.2%)	2002 (39.9%)
Medicaid	34 (22.0%)	17 (18.3%)	16 (26.7%)	189 (18.4%)	464 (18.6%)	38 (19.4%)	38 (14.8%)	128 (16.3%)	904 (18.0%)
No insurance	15 (9.8%)	† (4.3%)	11 (18.3%)	35 (3.4%)	113 (4.5%)	† (3.1%)	24 (9.2%)	74 (9.4%)	273 (5.4%)
Unknown	16 (10.4%)	12 (12.9%)	† (11.7%)	108 (10.6%)	241 (9.7%)	15 (7.7%)	24 (9.2%)	85 (10.8%)	497 (9.9%)

^a^. Includes all cases included here and other Hispanic; ^b^. *p*-value obtained from chi-square test for differences between detailed Hispanic ethnic groups not including US-born or foreign-born Mexican groups (i.e., Mexican, Puerto Rican, Cuban, Dominican, Central American, and South American); ^c^. Census tract poverty level is defined as the proportion of the population living below: very low (0% to < 5%), low (5% to < 10%), medium (10% to < 20%), high (20% to < 100%); ^d^. Includes n = 6 identified as non-binary. † Not reported due to fewer than 10 cases in group. Abbreviation: API: Asian/Pacific Islander; N: number; IQR: interquartile range; NOS: not otherwise specified; SEER: Surveillance, Epidemiology, and End Results Program.

**Table 3 cancers-15-02164-t003:** Age-Adjusted Incidence Rates and Incidence Rate Ratios of Lung Cancer by Detailed Race–Ethnicity and Sex. Florida, 2012–2018.

Race–Ethnicity	Cases (N)		Age-Adjusted Incidence Rate ^b^ (95% CI)		Incidence Rate Ratio (95% CI)	
	Male	Female ^a^	Male	Female ^a^	Male	Female ^a^
**NON-HISPANIC WHITE**	49,043	47,088	71.4 (70.8–72.1)	61.0 (60.4–61.6)	Reference	Reference
**NON-HISPANIC BLACK**	5553	4157	63.5 (61.8–65.3)	36.0 (34.9–37.1)	0.89 (0.86–0.92)	0.59 (0.57–0.61)
*US-born Black*	4852	3600	81.7 (79.3–84.2)	46.4 (44.8–48.0)	1.14 (1.11–1.18)	0.76 (0.73–0.79)
*Caribbean-born Black*	654	515	26.2 (24.1–28.4)	14.9 (13.6–16.3)	0.37 (0.34–0.40)	0.24 (0.22–0.27)
**NON-HISPANIC API**	512	518	25.5 (23.2–27.9)	18.8 (17.2–20.6)	0.36 (0.32–0.39)	0.31 (0.28–0.34)
**HISPANIC**	7038	5024	49.8 (48.6–51.0)	26.4 (25.7–27.1)	0.70 (0.68–0.71)	0.43 (0.42–0.45)
Mexican	276	153	36.8 (31.7–42.4)	16.8 (14.0–19.8)	0.52 (0.44–0.59)	0.27 (0.23–0.33)
*US-born Mexican*	172	93	59.8 (49.5–71.3)	23.5 (18.5–29.2)	0.84 (0.69–1.00)	0.39 (0.30–0.48)
*Foreign-born Mexican*	104	60	22.3 (17.4–28.2)	12.2 (9.1–16.0)	0.31 (0.24–0.39)	0.20 (0.15–0.26)
Puerto Rican	1224	1026	43.7 (41.2–46.4)	28.8 (27.1–30.7)	0.61 (0.58–0.65)	0.47 (0.44–0.50)
Cuban	4189	2491	65.6 (63.6–67.6)	31.7 (30.5–33.0)	0.92 (0.89–0.95)	0.52 (0.50–0.54)
Dominican	207	198	44.2 (37.8–51.2)	25.2 (21.8–29.1)	0.62 (0.53–0.72)	0.41 (0.36–0.48)
Central and South American	986	1047	30.8 (28.8–32.9)	20.8 (19.5–22.2)	0.43 (0.40–0.46)	0.34 (0.32–0.36)
Central American	208	258	24.1 (20.6–28.1)	16.3 (14.3–18.5)	0.34 (0.29–0.39)	0.27 (0.23–0.30)
South American	778	789	33.4 (31.0–36.0)	22.9 (21.3–24.6)	0.47 (0.43–0.50)	0.37 (0.35–0.40)

^a^. Includes n = 39 identified as non-binary; ^b^. Rates are annual, per 100,000, and age-adjusted to the U.S. 2000 Standard Population. Abbreviation: N: number; CI: Confidence Interval; US: United States of America; API: Asian/Pacific Islander.

## Data Availability

The data underlying this article are confidential public health records with personal identifiers that can only be released for specific use upon approvals from the Florida Department of Health Cancer Registry Program, Florida Department of Health Bureau of Vital Statistics, and the Florida Department of Health Institutional Review Board. These data are never available for public repository given the confidential information they contain. This study was approved by the University of Miami and the Florida Department of Health Institutional Review Boards. The datasets are available by request with required approvals from the Florida Department of Health Cancer Registry Program and Florida Department of Health Institutional Review Board. Applications for data request are available from the FCDS Webpage (https://fcds.med.miami.edu/inc/datarequest.shtml (accessed on 20 February 2023). Any published findings and conclusions are those of the authors and do not necessarily represent the official position of the Florida Department of Health.

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
