# Peer review of "Lung Cancer Incidence by Detailed Race–Ethnicity"

_cancers, 2023, doi:10.3390/cancers15072164_

Round 1

Reviewer 1 Report

This study investigated lung cancer incidence rates across detailed race-ethnicity in Florida, revealing significant disparities and highlighting the heterogeneity of the disease within racial-ethnic groups. The study provides important insight into the need for enhanced granularity in racial-ethnic stratification, identifying local populations that may benefit from targeted prevention programming and education, as well as enhanced lung cancer screening.

 While the study acknowledged its limitations, such as the lack of data on smoking by detailed race-ethnicity in the general population in Florida, it did not mention the potential impact of environmental factors, such as air pollution or occupational exposures, on lung cancer incidence. Exposure to air pollution has been identified as a risk factor for lung cancer, and studies have shown that people living in areas with high levels of air pollution have a higher risk of lung cancer, particularly when they smoke or have other risk factors.

One of the limitations associated with using age-adjusted rates in this study and similar studies is that they assume a constant age-specific rate across all age groups, which may not always be the case, especially when comparing Us born vs foreign born populations. Additionally, the choice of the standard population used to adjust for age can impact the results. Finally, age-adjustment alone may not be sufficient to fully account for confounding factors such as smoking status, occupation, and other environmental exposures. Also, providing additional details on the methods used for multiple imputation, along with a reference for readers to consult, would improve the paper's overall quality and favor reproducibility.

Overall, the study has important implications for public health surveillance and clinical professionals working with diverse populations, and it highlights the need for further research into the impact of environmental factors and the use of more refined approaches in the fight against lung cancer.

Reviewer 2 Report

In raw 43 the choosen word 'deadliest' is not the proper scientific word. There are several other cancer types which lethality is higher than the lung cancer lethality (e.g.: pancreatic cancer). However it is true that lung cancer has the highest mortality rates from cancer mortality in most of the high income countries. The mentioned word (deadliest) is not clear whether its meaning is 'lethality' or 'mortality'. Please clarify it by using the proper words like incidence, prevalence, mortality, morbidity, lethality.

In the later part of the article everything is clear.

In Table1 the smoking status is presented in percentage of known smoking status. However it would be also a very important information to know that how many percentage of the total group was never smoker vs. ever smoker vs. unknown. E.g. in male, Non-Hispanic White group the number of unkwon smoking status is higher than the number of never smokers (as in some other groups) however in non-hispanic Caribbean-born Black group No. of unkown status is only half of the No. of never smokers (which is a quite high subgroup with its 26% of known smoking status) or API or Hispanic group. Based on that it would not be correct to say that almost one fourth of lung cancer patients with API origin was never smoker compared the white Non-Hispanic group whaere it was less than 10%. And unfortunately this presentation of data suggest that misinformation instead of the real percantage of the total group. So I suggest to show that ratios as well.

In raw No 150 this articles suggests that the frequency of lung cancer is 13.8% or 14.7% or 23.3% etc. in Hispanic male never smokers. This is not the lung cancer proportion. This is the 'never smokers' proportion among lung cancer patients. It is a very important difference. The meaning of this artical is the following: if you are a hispanic never smoker male than your chance is 13.8% to develope lung cancer. It is not true. First of all this study never studied this correlation. Secondly the lung cancer incidence is much less frequent...

In raw 158 it is a good explanation '...proportions of private insurance...' Please mind the specific words and sentence structure.

Furthermore the problem is the same as in Table1. The unknown smoking status has a huge variety in ethnic subgroups. It would not be a true sentence that 'the proportion of never smokers among lung cancer patients vary from 13,8% to 23,0% in Hispanic population' as the unknown status subgroup should be also shown. This interpretation is misleading.

In raw 172 the expalanation of Figure1 is also a misinformation. Noone knows from this study any information about the percentage of lung cancer in never smokers. This study shows only the percentage of never smoking among lung cancer. And to tell the truth this study shows only the ratio between ever smokers and never smokers among lung cancer patient and not the real percentage of never smokers as the unknown status is neglected despite its considerable size.

Starting from raw 175 the interpretation of the date is chaotic.

Table3 and interpretation of Table3 is alright however it is not clear whether these data (incidence rates) are standarized rates or just based on the native numbers (crude rates). If these rates are not age-standardised rates than the comparison is absloutely fals. It is also a wellknown information that the age structure of the ethnic groups has huge variety. That is why the comparison can not be presented without standardisation.

This article needs a very accurate statistical review. The interpretations of the results are consistently wrong.

Discussion sould be also partly rewrited based on the appropriate statistical calculations.

I would be happy if I could read some more information about those registries which contain the patients' race-ethnicity. In  several European countries the registration of race-ethnicity in medical documentation is illegal. I know that in some other countries it is possible to collect these data but I would like to understand the method of this data collection. Are people generally asked about their race-ethnicity during the medical examination? Is it generraly part of the medical history or just specifically in some special diseases? 

Reviewer 3 Report

Thanks for the opportunity to review this manuscript. This is a study conducted in racial-ethnic populations using lung cancer (LC) data from the Florida registry between 2012 and 2018 period (120,550 LC cases analyzed). The objective of the study by Cranford H. et al was to assess LC heterogeneity among race-ethnicity groups and LC cancer cases from the Florida cancer registry, 2012-2018. The authors showed differences in the LC incidence according to the ethnic group.

The study sample is very large, including men and women from four races-ethnicity and minority populations from different areas, thus allowing for sufficient geographical and cultural variability sociodemographic characteristics and habits such as tobacco consumption, very important for LC. Results are interesting and important to inform about the disparities in LC incidence in racial-ethnic populations for public health surveillance and professionals that working with diverse US populations. It is a very important topic, but there are major concerns arose during the review of the manuscript, several of which are highlighted below.

11.  The authors should describe the Cancer Registry and its strengths. For example, How long has the Cancer Registry been collecting and reporting cancer data? What are the data covered about of the by the registry? How are cases registered (hospital registry, primary care)? Data quality check was assessed?

22.  The authors present mainly descriptive results, and do not intend to delve into more analytical results such as analysis of incidence trends, or incidence rates year by period or year (2012 to 2018).

33.  The LC incidence can be analyzed according to each age group, and if possible according to histology and stage.

44.  It would have been very interesting to see the LC incidence of each race-Ethnicity according to smoking status separately in men and women.

55.  The authors should describe how the smoking habit was collected and whether it was through the questionnaire.

66.  In Table 3, the number of cases should be included.

77.  What was the percentage of cases that had confirmed histological? This should be included and controlled in the analysis.

88.  I suggest analyzing the incidence LC according to cases with microscopic verification and cases without microscopic verification.

99.  The authors must indicate which are the ‘Others Specified’ in the distribution by histology (Table 1 and Table 2).

110.  I would add to the limitations that there is a lack of information on clinical variables such as histology and stage.

111.  Figure 1, I could not find.

Round 2

Reviewer 2 Report

I accept the Authors' decision as not presenting the smoking status of participants. Although I hoped that the "unknown smoking status" group would also be included in the analysis. No problem, excellent research in this form as well and now I don't see any confusing interpretation or analysis in it. Congratulation for this really uniq study.

Reviewer 3 Report

It would have been more interesting, in addition, the descriptive analysis, to have been able to see the trend or analysis during study periods. Therefore, the results lose their relevance as incidence study. With the exception of this concern, the authors have addressed to my other points raised satisfactorily.